# Assessing the Social Cohesion of a Translocated Pride of White Lions Integrated with Wild Tawny Lions in South Africa, Using Social Network Analysis

**DOI:** 10.3390/ani12151985

**Published:** 2022-08-05

**Authors:** Jason A. Turner, Hans de Iongh, Emma J. Dunston-Clarke

**Affiliations:** 1Institute of Cultural Anthropology & Developmental Sociology, Leiden University, 2300 Leiden, The Netherlands; 2Institute of Environmental Sciences, Leiden University, 2300 Leiden, The Netherlands; 3School of Veterinary Medicine, Centre for Animal Health and Welfare, Food Futures Institute, Murdoch University, Murdoch, WA 6150, Australia

**Keywords:** African lion, *Panthera leo melanochaita*, white lion, conservation, translocation, behaviour, social interactions

## Abstract

**Simple Summary:**

Lions in South Africa are protected in national parks and smaller fenced private wildlife reserves. Population sizes and genetics within fenced reserves are managed through moving lions between reserves. One component of successful lion movements is their ability to form prides post release, which is influenced by the strength of social bonds and interactions between individual lions. White lions are a natural colour variant of the African lion, occurring within certain tawny lion prides in the Greater Kruger National Park, South Africa. Human impact, including trophy hunting, led to the removal of white lions in the wild until their reintroduction in 2006. The social behaviour of a pride consisting of captive-origin white and wild tawny lions was compared to captive-origin and wild tawny lion prides, with similarities and differences in the social behaviour of prides found. The study concluded that the pride of white and tawny lions was more strongly bonded than either the captive-origin or wild tawny prides. This suggests that a constructed pride of captive-origin white and wild tawny lions can successfully form a socially functional lion pride and indicates that white lions are capable of surviving in the wild in the absence of negative human impact.

**Abstract:**

In South Africa, lions are protected in national parks and smaller fenced reserves. Translocating lions between fenced reserves, whilst necessary to maintain genetic diversity, is disruptive and can impact survivorship and pride cohesion. Critical to translocation success is pride cohesion. White lions are a natural colour variant occurring in the Greater Kruger Park Region, where anthropogenic threats eliminated this population until reintroduction in 2006. Through social network analysis (SNA), the sociality of a released pride of captive-origin white and wild tawny lions was compared to two captive-origin and wild prides of tawny lions. Social interactions and pride dynamics were recorded for each pride. For all prides, cubs and subadults were central to the play network, while adults received the most social interactions. White and wild tawny adult males initiated more social interactions than captive-origin tawny males, whilst a keystone adult female was identified in each pride. For the constructed pride, social interactions were more evenly distributed, suggesting a high level of connectedness and cohesion. This is the first study to demonstrate that captive-origin white and wild tawny lions can form a socially functional pride, suggesting that white lions would survive in the wild in the absence of anthropogenic threats.

## 1. Introduction

The conservation status of the African lion (*Panthera leo*) in Southern Africa is listed as ‘Vulnerable’ [1]. Significant population declines are due to prey depletion, loss of habitat, human–wildlife conflict, climate change, inbreeding depression, and disease [1,2,3,4,5,6]. In South Africa, lions went extinct throughout much of their former range during the early 20th century [7], with approximately 1875 lions located in three populations [8,9,10]. According to Bauer et al. (2015) [5] and Packer et al. (2013) [11] this species’ survival in southern Africa may increasingly depend on populations in fenced, intensively managed reserves.

The translocation of lions between fenced reserves in South Africa is required regularly to maintain genetic diversity and population management [12]. Such translocations between small, fenced reserves (<1000 km^2^) have occurred from 1990 to 2013, involving approximately 800 lions [13], with this population now managed as a metapopulation [12,14]. The use of wild individuals in translocations is preferred due to the higher failure rate reported for captive-origin animals [15,16,17] and according to Hunter et al. (2013) [18], this also applies specifically to lion translocations. However, captive-born lions may become necessary when no suitable wild individuals remain, or when the surviving wild population is no longer viable [19,20,21]. Known as ex situ reintroduction [22], the release of captive-origin individuals has been performed successfully for many species, including golden lion tamarin [23], red wolf [24], California condor [25], Arabian oryx [26], and the black-footed ferret [27], proving to be an important tool that has prevented the wild extinction of some of these species.

The white lion is a natural colour variant, or leucistic form, of the southern subspecies of the African lion (*Panthera leo melanochaita*) that had been prevalent in the Greater Timbavati Region [28,29,30] and Central Kruger Park Region [31,32,33] of South Africa since 1938 [34]. Being extremely rare in the wild, white lions and many tawny lions that had the recessive gene were removed from the wild for captive breeding and hunting programmes, zoos and circuses worldwide [29,30,34]. The combination of lion culling in Central Kruger National Park [32], in addition to trophy hunting and illegal removal from the Greater Timbavati Region into captive breeding centres [29,30,34,35], resulted in white lions being absent from their natural region up until 2006. Even though white lion cubs were born into the Greater Timbavati Region in 2006 to 2009, 2011, 2014, 2015, 2018, and 2019, and in Central Kruger National Park in 2015 and 2016, only three of the seventeen cubs had survived at the time of the present study due to anthropogenic impact. The specific impacts were illegal removal to breeding centres, the continued lion trophy hunting of pride males, which caused infanticide, and high-impact ecotourism leading to undue stress on lionesses with young cubs during regular viewing by tourist or lodge vehicles [36,37,38]. In addition to the anthropogenic impact, the natural high mortality rate of 50% of lion cubs within the first year [39] would also have contributed to the low survival rate of white lion cubs between 2006 and 2019. Adult white lions were therefore reintroduced to a managed free-roaming wildlife area in 2006 by the Global White Lion Protection Trust and became self-sufficient and successfully raised offspring. Turner et al. (2015) [35] showed that there was no difference in the hunting success of the two reintroduced white lion groups compared to wild tawny lions in the same study area, Madjuma Lion Reserve (MLR), Karongwe Game Reserve (KGR), Welgevonden Game Reserve (WGR), Makalali Game Reserve (MGR) and the Associated Private Nature Reserves (APNR) in South Africa. The home range behaviour of white lions was also shown to be similar to that of wild tawny lions by Turner et al. (in prep.) [40].

Although the overall success of lion reintroduction is the raising of offspring and population increase post-release, a key factor in determining the short-term success of either lion translocation or reintroduction is group cohesion [41,42,43]. Lack of social cohesion can increase mortality and dispersal post-release [41], and reduce reproductive success, ultimately resulting in post-release failure [42]. A lion pride is a cohesive social unit composed of a core of related females, their cubs and one or more males [44]. Naturally, lion prides are composed of between 2 and 9 adult females (range of 1–21), their dependent cubs and subadults, and between 2 and 6 adult males (range of 2–9) [45], with the average pride consisting of 15 individuals but can range from 4 to 37 [44]. Prides exist as a fission–fusion society, with pride members regularly splitting into smaller groups and then reforming, seldom all being grouped together [46]. Social cohesion and cooperation are fundamental to the success of the pride, with benefits including territory defence to increased reproductive success [45], coordinated hunting [47] and communal nurturing of young [48]. Social greetings and allogrooming are regarded as critical in the maintenance of social bonds between lions in captive [48] and wild prides, whereas play behaviour is mainly carried out by cubs as part of their development [44]. Social network analysis (SNA) has previously been found as useful in assessing and quantifying animal social structures, at both a group and individual level [49,50]. Abell et al. (2013a) [20] provided the first SNA on lions, focusing on a captive-origin lion pride with wild-born cubs, identifying that the roles of individuals, sexes and age groups varied and that the pride was socially cohesive. Dunston et al. (2017) [51] conducted the first SNA to compare the sociality of two captive-origin lion prides with that of a wild lion pride, finding that the captive-origin prides were socially cohesive, displaying behaviour and interactions similar to that of the wild pride. SNA has also been used more recently by Mzileni et al. (2019) [52] to study horizontal disease transmission in wild lions in Kruger National Park, by looking at the behavioural interfaces of lions. However, SNA has never been used to determine whether reintroduced captive-origin (white) lions integrated into a pride can form a socially cohesive pride.

Our study used SNA to determine whether a pride consisting of two white lion males originating from captive origin had formed a socially cohesive pride with two translocated wild tawny lionesses and their offspring. This was achieved by comparing our pride to previously studied captive-origin and wild prides. Notably, the two tawny lionesses were unrelated, originating from two different wildlife reserves in South Africa, further complicating the development of a cohesive pride. Establishing the sociality of reintroduced lion prides is an important aspect of determining post-release success and will inform future translocation and reintroduction programs, which may prove critical to the metapopulation management approach. This is particularly pertinent due to the increasing impact of anthropogenic threats to lions including poaching, trophy hunting, and human–lion conflict. 

## 2. Materials and Methods

### 2.1. Study Site

This study was conducted at the Tula-Tsau Conservation Area (7 km^2^), a fenced wildlife area in the Kruger to Canyons Biosphere, Limpopo Province, South Africa. The Tula-Tsau Conservation Area is part of an important buffer area (Greater Kruger Environmental Protection Zone (GKEPZ)) between the Kruger National Park, rural communities, and the semi-urban town of Hoedspruit (Figure 1).

The Tula-Tsau Conservation Area is a managed wildlife area that consists of a mixed vegetation type that is classified as Arid Lowveld [53], with an undulating landscape consisting of plains, woodlands, thickets, and riverine vegetation. A wide variety of large mammalian prey species occur, including the blue wildebeest *Connochaetes taurinus*, Burchell’s zebra *Equus quagga*, Greater kudu *Tragelaphus strepsiceros*, common warthog *Phacochoerus africanus*, and impala *Aepyceros melampus*. Black-backed jackals, *Canis mesomelas* and caracals *Caracal caracal* are commonly seen in the area, and both leopards *Panthera pardus* and spotted hyaenas *Crocuta crocuta*, while seen intermittently are not established in the reserve. Prides of wild lions occur on the two neighbouring private nature reserves.

Since there were no adult white lions in the wild in 2004, a founder group of white lions of captive origin was translocated to the Tula-Tsau Conservation Area—a 17.2 km^2^ fenced wildlife area bordering the Timbavati Private Nature Reserve in the Kruger to Canyons Biosphere. In 2006, the founder pride of white lions was successfully translocated to the Tula-Tsau Conservation Area and became self-sufficient, hunting sufficient natural prey to satisfy the needs of the pride without requiring any human intervention [35]. In 2013, the two male offspring of the founder pride were subsequently integrated with translocated wild tawny lions at the Tula-Tsau Conservation Area to form the Tsau pride. 

The Tsau pride comprised two 8-year-old sibling adult white lion males that were integrated with two unrelated wild tawny lionesses translocated from other reserves in 2013 and 2015, respectively. The first lioness (6 years old) originated from the 111, 000 ha Tswalu Kalahari Reserve (TKR), and the second lioness (12 years old) came from the 12,000 ha Kapama Private Nature Reserve in the Limpopo Province of South Africa. In December 2018, a single litter of three tawny lion cubs was born. 

The lions were fitted with VHF radio tracking collars and monitored from a research vehicle twice daily, at dawn and dusk, from 7 April to 10 May 2019 for between 1–5 h per session. At each monitoring session, pride location, group composition and social interactions were recorded. Group composition was recorded at the start and end of each monitoring session, or when a change in the composition took place due to a pride member leaving or joining, with each lion being recorded as present or absent. Social interactions were recorded via all-occurrence sampling, with interaction type (greet, groom, play and aggression, as previously described by Schaller 1972 [44]), the individual who initiated and received the interaction, and whether it was accepted, ignored, or rejected being recorded. A social interaction event was deemed to have ended after 1 min without interaction. In the case of multiple interactions being observed, only the initial interaction type was recorded, avoiding pseudo replication. 

The focal lion group for this study was the Tsau pride. The social behaviour and cohesion of this pride was compared to previously studied captive-origin tawny and wild prides, detailed in Dunston et al. (2017) [51]. The 2 captive-origin prides were situated in Livingstone, Zambia (Dambwa pride) and Gweru, Zimbabwe (Ngamo pride). The wild pride was located at the Greater Makalali Private Game Reserve (GMPGR) (Makhutswi pride) and was studied twice, in 2014 and 2015, respectively, where the pride composition changed, and is therefore referred to as the Makhutswi 1 and Makhutswi 2 prides. All prides occurred in fenced reserves within different countries in southern Africa. The pride composition for the 5 prides is detailed in Table 1.

### 2.2. Statistical Analysis

Statistical analysis of the social interaction data was based on the approach used by Abell et al. (2013a) [20] and Dunston et al. (2017) [51]. All of the social interaction data were standardized for each pride on a pairwise and hourly basis. The prides were observed on 66, 46, 20, 26 and 30 separate occasions for the Tsau, Ngamo, Dambwa, and Makhutswi 1 and 2 prides, respectively. The total observation time (hours) and number of social interactions recorded for each pride was: the Tsau pride (93 h; 622 interactions), the Ngamo pride (98 h; 667 interactions), the Dambwa pride (67 h; 841 interactions), the Makhutswi 1 pride (62 h; 162 interactions), and the Makhutswi 2 pride (30 h; 43 interactions). Asymmetrical (directional), weighted matrices were calculated for greeting, social grooming, play, aggression, and all social interaction types and standardized by dividing the number of interactions collected per pair of lions by the total number of hours each pair was observed together per pride. A symmetrical matrix was compiled from pride composition data, and a modified ratio index (Abell et al. 2013 a) [20] was used to show individual association value within the pride. 

Social interactions and pride compositions were analysed at an individual (degree and betweenness centrality) and pride level (density and clique) for each of the five prides. The density, degree (indegree and outdegree), betweenness centrality, and clique groups for each network per pride were calculated using the SNA statistical program UCINET, version 6.543 [54]. Symmetrical matrices were generated prior to calculating betweenness centrality and cliques for each network per pride. Normalized indegree and outdegree values were generated using UCINET, and a Spearman’s correlation was conducted to test for dependence within and between networks for each pride. Kendall’s tau correlation was used to examine any relationships for betweenness (centrality) and degree (indegree and outdegree). For a detailed explanation of density, degree, betweenness centrality and cliques, please refer to Abell et al. (2013a) [20], Sih et al. (2009) [50], Dunston et al. (2017) [51], and Wey et al. (2008) [55].

The sociograms and clique figures for each network were generated using NETDRAW, version 2.1476 [51,54]. The association between social interaction matrices and age, gender, kinship, pride composition, and a random network was assessed by means of a Mantel test using SOCPROG version 2.4 [56]. The random network was generated in UCINET for each pride. 

## 3. Results

Analyses showed that social interactions at the group and individual level within each pride differed between networks, indicated by density, degree, and betweenness centrality. There was no significant association between any of the interaction networks with a random network, indicating that interactions were non-randomly distributed within each pride. 

Tsau pride density values for all social and greet matrices were similar to both captive-origin prides (Ngamo and Dambwa), which were all higher than the wild Makhutswi prides, indicating that these prides were highly connected (Table 2). 

A correlation was found between all social and greeting matrices for all prides, with the majority of pride members being connected for each of these networks (Figure 2 and Figure 3). For the Tsau pride, all social and greeting interactions between some individuals were lower. This is reflected in the clique matrices for both networks, where adult female C is associated with one clique and adult male M to the second clique (Figure 2 and Figure 3). Interactions between individual lions were also absent within the Ngamo and both Makhutswi prides, resulting in 2 and 4, 5 and 5, and 8 and 1 cliques found for all social and greeting matrices for these prides, respectively. The Dambwa pride had only one clique for these matrices and had all lions involved in the clique. For four prides, a cub (Tsau = Cu1, Dambwa = LE1) or subadult (Ngamo = AS5, Makhutswi 1 = SA) was most likely to initiate social interactions (Figure 4), while an adult lion was observed to be the recipient of interactions (Tsau = T, Ngamo = AS, Dambwa = ZU, and Makhutswi = LB), and least likely to initiate interactions (Ngamo = MI, Dambwa = ZU, and Makhutswi 1 = MID). In contrast, for the Tsau pride, a male cub (Cu1) was observed to receive a low number of interactions, whilst another male cub (Cu2) received the most social interactions and initiated a low number of interactions (Figure 4 and Figure 5). Similarly, the Makhutswi 2 pride had a cub (Cub2) that initiated and received the second highest number of social interactions, but the highest number of social interactions was initiated and received by an adult male (XI) (Figure 4 and Figure 5).

The Tsau pride greet matrix showed that interactions between adult males M and Z, between male Cub1 and Z, and between the three cubs and adult female T were strongest. Strong connections between adult and subadult pride males were also observed for the Ngamo, Dambwa and Makhutswi 2 prides, while connections between pride females and their cubs were found for all prides (Figure 3). A positive association between the greeting and all social interactions for all five prides was found: Tsau (_r_M = 0.5949, *p* = 0.0003), Ngamo (_r_M = 0.8828, *p* = 1.000 × 10^−13^), Dambwa (_r_M = 0.9276, *p* = 1.000 × 10^−13^), Makhutswi 1 (_r_M = 0.9201, *p* = 1.000 × 10^−13^), and Makhutswi 2 prides (_r_M = 0.7077, *p* = 2.0006 × 10^−13^). Spearman’s correlation found no association between greet indegree and greet outdegree for the Tsau (π = −0.018, *p* = 0.869), Makhutswi 1 (π = 0.536, *p* = 0.329) and Makhutswi 2 prides (π = 0.719, *p* = 0.432), but was negatively associated for the Ngamo (π = 0.627, *p* = 0.039) and Dambwa (π = 0.785, *p* = 0.003) prides. Kendall’s Tau analysis found no association between greet centrality and any of the other indegree or outdegree social interactions for the Tsau pride (*p* > 0.05) (Appendix A: Table A2).

Across all prides, subadults were most likely to receive grooming interactions (Tsau = Cu3, Ngamo = AS4, Dambwa = LE1, Makhutswi 1 = DA, Makhutswi 2 = Cub2) (Figure 5 and Figure 6). For each pride, an adult female and pride subadults had the highest betweenness centrality for the grooming network (Figure 7), with cubs/sub-adults most likely to receive and adult females or adult males (when involved) to initiate interactions (Figure 4 and Figure 5). Pride males, in all prides except the Tsau pride, were the least likely to receive grooming interactions, and for the Ngamo and Makhutswi 1 prides, initiated the least number of this interaction type. In contrast, for the Makhutswi 2 pride, the pride male (XI) initiated the most grooming interactions. For the Tsau pride, adult female C and cubs Cu1 and Cu2 received a lower number of grooming interactions compared to the pride males. A positive association between grooming and all social interactions for four of the prides was found: Tsau (_r_M = 0.4672, *p* = 0.0393), Ngamo (_r_M = 0.2896, *p* = 0.0114), Makhutswi 1 (_r_M = 0.1944, *p* = 0.0372), and Makhutswi 2 (_r_M = 0.6750, *p* = 0.0030). No association between grooming indegree and grooming outdegree nor between grooming centrality and any of the other indegree or outdegree social interactions was observed (*p* > 005) (Appendix A: Table A1 and Table A2).

Across all play matrices, cubs/sub-adults were both central to and most likely to initiate and receive play interactions (Figure 6), with some to most adults of the pride involved to a lesser extent (Figure 4 and Figure 5). Prides that consisted of cubs (Tsau, Dambwa and Makhutswi 2 pride in particular) had more pride members involved in this network (Figure 5). For all prides, cubs and subadults initiated and received the greatest number of play interactions (Figure 8 and Figure 9), which was reflected in these lions having high betweenness centrality values for this network. For the Tsau pride, the cubs who initiated and received the most play interactions also initiated and received the most cumulative social interactions, respectively (_r_M = 0.821, *p* = 0.034, _r_M = 0.786, *p* = 0.048). For the Dambwa pride, individuals who received the most play interactions also initiated the most social (_r_M = 0.741, *p* = 0.006), greeting (π = 0.581, *p* = 0.047), and grooming interactions (π = 0.680, *p* = 0.038). The Tsau pride cubs that received the most play interactions also received the greatest number of all social interactions (π = 0.821, *p* = 0.034). Positive correlations between play indegree and outdegree were found for the Dambwa (π = 0.680, *p* = 0.015) and Makhutswi 1 pride (π = 0.764, *p* = 0.004), while four of the five prides had a positive association between play centrality and play outdegree (Tsau: π = 0.926, *p* = 0.005; Ngamo: π = 0.607, *p* = 0.024; Dambwa: π = 0.667, *p* = 0.003; Makhutswi 1: π = 0.620, *p* = 0.019). The Tsau pride also had a positive association between play indegree and all social centrality (π = 0.700, *p* = 0.042) and a positive association between play and all social interactions (_r_M = 0.8255, *p* = 0.001).

For the aggression matrix, the Tsau pride was the only pride to have all pride members involved and mostly connected in this social network, but still had four cliques, indicating that interactions were not distributed evenly (Figure 6). Most aggression interactions were acts of discipline directed towards the male Cub1 from both adult males and adult female T, with aggression also observed as a territorial or defensive display between the adult males and between the adult males and adult female T. Aggression interactions from adults, particularly parents, towards pride cubs/sub-adults were observed for all prides, while some interactions between adult pride members were also observed. A positive association was found between aggression and the cumulative social interactions for the Makhutswi 1 (_r_M = 0.2353, *p* = 0.0341) and Makhutswi 2 prides (_r_M = 0.8035, *p* = 2.0006 × 10^−13^). No association between aggression indegree and aggression outdegree was found for the Tsau pride (π = 0.536, *p* = 0.498), nor any association between aggression centrality and any indegree or outdegree networks (*p* > 0.05) (Appendix A: Table A2), indicating that the lion that was central to this network (adult lioness T) did not receive or initiate the most social interactions for those networks.

Social interaction networks for all prides were not significantly associated with the random network, indicating that the interactions were non-randomly distributed within each of the prides (Appendix A: Table A3, Table A4, Table A5, Table A6 and Table A7). The degree to which individuals initiated and received interactions within each pride indicated their role within an interaction (Figure 8 and Figure 9). There was no significant correlation between initiated and received social interactions for the Tsau pride for any network (*p* > 0.05) (Appendix A: Table A1), indicating that social interactions were more evenly split between pride members and suggesting a high level of connectedness. In contrast, for the Dambwa and Makhutswi prides, all social indegrees and outdegrees were negatively correlated (π = −0.727, *p* = 0.007; π = 0.729, *p* = 0.002), and for the Makhutswi 2, pride all social indegrees and outdegrees were positively correlated (π = 0.712, *p* = 0.009), indicating that social interactions for these networks were less evenly split between pride members. For the Tsau, Dambwa and Makhutswi 2 prides, pride members who initiated the most cumulative social interactions were also found to initiate the most greetings (π = 0.955, *p* < 0.001, π = 0.767, *p* = 0.000, and π = 0.855, *p* < 0.001). The test of association for the Tsau pride found no correlation between the centrality for greeting, grooming or aggression and any outdegree or indegree social interactions (*p* > 0.05) (Appendix A: Table A2), indicating that the lions who were central to these networks did not receive or initiate the most social interactions for those networks. A positive association was found for the Tsau pride between the grooming and greeting social interactions (_r_M = 0.4297, *p* = 0.0384) and between grooming and all social interactions (π = 0.46715, *p* = 0.0393).

The Mantel test results per pride indicated whether there was an association between the social interaction networks and full siblings, age, sex, and pride composition. Full siblings and age were not associated with cumulative social interactions for the Tsau, Ngamo, or Makhutswi 2 prides (Appendix A: Table A3, Table A4 and Table A5). A positive association was found between social interactions and full siblings (_r_M = 0.1329, *p* = 0.039, and _r_M = 0.2096, *p* = 0.0189) and age (_r_M = 0.1445, *p* = 0.0218, and _r_M = 0.2096, *p* = 0.0188) for the Dambwa and the Makhutswi 1 prides. For the Tsau, Dambwa and Makhutswi 2 prides, no significant association was found between any of the social networks with sex (Appendix A: Table A3, Table A4 and Table A5) or for age for the Tsau, Ngamo, and Makhutswi 2 prides (Appendix A: Table A3, Table A4 and Table A6).

The Tsau and Makhutswi 1 prides had a positive association between pride composition and grooming (_r_M = 0.4658, *p* = 0.0245, and _r_M = 0.2188, *p* = 0.0230), play (_r_M = 0.3721, *p* = 0.0191, and _r_M = 0.2817, *p* = 0.0338), and all social (_r_M = 0.4965, *p* = 0.0027, and _r_M = 0.2797, *p* = 0.0385) interactions. For the Ngamo pride, no significant association was found between pride composition and any social networks, full siblings, sex, or age (Appendix A: Table A4). For the Dambwa and Makhutswi 1 prides, a positive association was found for pride composition and full siblings (_r_M = 0.7031, *p* = 1.0003 × 10^−13^, and _r_M = 0.4468, *p* = 1.000 × 10^−13^) and for pride composition and age (_r_M = 0.5571, *p* = 0.0021, and _r_M = 0.4468, *p* = 1.000 × 10^−13^). A positive association was found between pride composition and full siblings for the Makhutswi 2 pride (_r_M = 0.5369, *p* = 0.0008).

## 4. Discussion

Critical to determining short-term successful reintroduction of a gregarious species is group cohesion. This study provided the first assessment of sociality of a constructed pride of African lions consisting of captive-origin white lions and wild tawny lionesses. By comparing the sociality of this pride (Tsau pride) to two previously studied captive-origin and two wild prides, we identified similarities and differences in social behaviour at an individual and pride level. Networks across all prides were found to be non-random, indicating that individuals chose to interact socially with other pride members. Social interactions were more evenly split between pride members in the Tsau pride compared to both the captive-origin and wild prides, suggesting a high level of connectedness and pride cohesion. The Tsau pride was identified to exhibit natural and expected social behaviours, leading to the identification of a keystone lioness and subadults/cubs to be central to the play network. Interestingly, Tsau pride males had a lesser role in the grooming network, while all pride members were involved in the aggression network (as a territorial defensive, or disciplinary display), compared to all other prides.

The Tsau pride was found to display natural social behaviours, with similarities to both the captive-origin and wild prides. All social and greeting interactions for all prides indicated that most of the pride members were connected and showed social cohesion for those networks. The Tsau pride was found to be highly connected due to most pride members being involved in all social networks. This was indicated by the high density values for all social networks for the pride, which were between the values found for the captive-origin prides and the wild prides. For all prides, there was a strong connection between adult and subadult pride males and between pride females and their cubs. This is typical of wild lion prides and is fundamental to the role that males and females play in prides and overall pride cohesion [44]. Similar to the captive-origin and wild prides, the cubs and subadults of the Tsau pride were central to, and most likely to initiate and receive, play interactions. Play behaviour is typical of cubs, as they hone the skills necessary to hunt prey successfully and defend themselves from attack by conspecifics or other carnivores [44], indicating that the cubs from the Tsau Pride were behaving naturally as wild cubs do. The social behaviour observed for both captive-origin white males and their offspring show no indications of being impaired by a captive origin or human impact. For each pride, an adult female and pride subadults had the highest betweenness centrality for the groom network. This was not unexpected given that it is regarded as a behaviour that strengthens bonds between mothers and their offspring and between adult females, with pride males largely absent from allogrooming [44,51]. Each of the five prides had a keystone adult female, with three of the four females having offspring within their prides and all being of a similar age (6–9 years old). Keystone females were involved in the majority of the social networks and played a critical role in connecting peripheral members to the rest of the pride. The role of a keystone female is critical to pride structure and stability within wild prides, but in their study, Dunston et al. (2017) [51] concluded that a captive history does not prevent this role from being fulfilled. Our study on the Tsau pride has shown that the integration of captive-origin white lion males with wild translocated tawny females did not inhibit the ability of one of the lionesses to establish themself as a keystone female. Importantly, the fact that the two adult lionesses in the Tsau pride were unrelated and translocated from different reserves yet were highly connected to each other and formed a socially cohesive pride indicates that such human management and intervention can be successful.

The Tsau pride expressed some social behaviours that differed to those previously observed for the captive-origin and wild prides. Tsau pride males received more grooming interactions than all other pride males. The most probable reason is that the Tsau pride males were brothers and therefore very closely bonded, whereas the captive-origin prides only had one pride male and the wild pride males were related (uncle/nephew) but were not siblings. For the Tsau and Makhutswi 2 prides, a pride male initiated a high number of the grooming interactions, which is probably due to the presence of cubs in both these prides, which often results in more frequent social interaction. Interestingly, despite the social cohesion of the Tsau pride, a greater number of aggressive interactions were observed compared to the other four prides. This mostly involved defensive or territorial behaviour rather than damage-causing behaviour between the four adult members of the Tsau pride and took place either when the adult lionesses were in oestrus, leading to more male to male and female to male aggression, or when the pride were feeding on a kill. These incidences of aggression are consistent with natural pride dynamics, especially during feeding and where a male coalition is present [44], indicating that the social behaviour of the Tsau pride was not negatively influenced by a captive origin or human impact.

The roles that individuals, sexes and age groups have on social interactions and pride social cohesion varied between prides. The Tsau pride was the only pride that had no correlation between full siblings for any of the social networks. However, this was unsurprising considering that the only full siblings in this pride were the adult males, and other than the groom network, the number of interactions between them was not sufficient to show a significant correlation. Adult males in wild prides are known to have a low level of overall social interaction [44,57]. The Tsau pride lionesses were unrelated and therefore not full siblings. Social interactions within the Tsau pride were also not significantly associated with the sex of the lions; however, a sex bias was observed, with lionesses directing these interactions to other lionesses, and a high number of greeting interactions taking place between the adult males. The adult males being siblings with a long period of association (8 years), and the natural tendency of lionesses to interact socially with each other [44], is the likely explanation for this sex bias for the Tsau pride. The captive origin of the Tsau males is probably another reason. A similar sex preference is common in captive lions that are kept in zoos [48]. Although a sex bias was also observed for the Ngamo pride, it was only for the lionesses. These findings support our other observations that there is a high level of connectedness and pride cohesion within the Tsau pride despite a captive origin and history of human impact through translocation.

Pride composition was influenced by various factors (social interactions, full siblings, sex and age) that differed between prides, suggesting that the origin of the pride (wild or captive origin) is an unlikely influence. The pride composition for the Tsau pride was, however, more similar to that of the wild Makhutswi 1 pride, being positively associated with groom, play, and all social interactions, suggesting that the social interactions and bonds between the white lion males and the rest of the pride was similar to that of a wild pride. This also implies that the integration of white lion males with tawny lionesses has been successful in forming a socially cohesive pride that behaves as a wild pride.

Overall, the Tsau pride was more highly connected socially than either the captive-origin or wild prides. The captive-origin Ngamo and Dambwa prides were in small reserves that had no other lions in the area and did not have to spend time being separated for hunting and territory maintenance behaviours, allowing more time to conduct social behaviour. In contrast, the Tsau pride did have to hunt for themselves and respond to territorial threats on the reserve boundaries, a natural behaviour for wild prides, and was therefore more similar to the Makhutswi prides. The wild prides had fewer social interactions compared to the Tsau pride due to a greater level of pride dispersal and a larger territory size. Adult pride members were often absent from the wild Makhutswi prides, compared to the integrated Tsau pride and the captive-origin Ngamo and Dambwa prides. Dispersal of pride members is not uncommon, with lion prides living in a fission-fusion society [44,45,47], and adult males regularly being away from their prides is a natural occurrence [39,44,49]. The wild male lions were often absent when spending time with another pride within their territory and when away patrolling the territory. The higher level of social interaction and presence of male lions observed with the Tsau, Dambwa and Ngamo prides could be accounted for by the difference in vegetation type and smaller reserve size. In dense vegetation, visibility is inhibited, requiring the pride, and particularly males, to be more mobile and active. In Kruger Park, pride males occur more regularly with their pride in open habitat compared to woodland habitat [58]. The territory of the Makhutswi prides was covered with both open and closed woodland, and vegetation is therefore a possible factor affecting the presence and absence of the Makhutswi males with the two prides. By the same token, the territory of the Tsau pride included many open plain and open woodland areas, a likely factor leading to the males spending more time with the pride. Additionally, the smaller territory size of the Tsau, Ngamo and Dambwa prides led to less fission–fusion changes within these prides, with the result that a greater number of pride members were present more frequently. The Tsau pride was located in a reserve four times the size of the Ngamo pride, and double the size for the Dambwa pride, but a fraction of the size for the Makhutswi prides (Table 1). Although lion territory sizes vary significantly depending on prey availability and pride size, and the fenced reserves of the Tsau, Ngamo and Dambwa prides are significantly smaller than the average territory sizes of wild prides; with a range of 50 to 7400 km^2^ [59,60,61,62], and a mean of 56 km^2^ (range of 15 to 219 km^2^) [45], nearly 20% of the wild lion population in South Africa is protected within 49 smaller fenced reserves, with several of them being significantly smaller than the reserve for the Makhutswi pride: Mabula Game Reserve (16.5 km^2^), Thanda Private Game Reserve (70 km^2^), Karongwe Game Reserve (79 km^2^), Thornybush Nature Reserve (116 km^2^), and Shamwari Game Reserve (139 km^2^) [12]. Lion prides within these smaller fenced reserves are now being managed as a metapopulation [62,63,64], based on natural metapopulation structures (Miller et al. 2014), which requires lion translocation to maintain genetic diversity and to prevent overpopulation [12]. The success of lion reintroduction and translocation has been linked to group cohesion, with post-release failure often being due to lower fecundity and a greater incidence of dispersal from the release site [42]. Lion prides located in smaller reserves with more open habitats therefore seem to have a higher level of social interaction and cohesion and may account for translocation success in South Africa. The Tsau pride showed social cohesion and pride stability, despite the varying origins of its members. This cautiously suggests that captive-origin and wild lions can be successfully integrated and reintroduced to smaller reserves if such management is required under metapopulation management.

The present study had several important findings, but also limitations and factors that need to be considered. As stated by Dunston et al. (2017) [51], the captive origin of pride members needs to be considered as having a possible impact on the social interactions of these prides. For the Tsau pride the captive origin of the white lion males may have led them to be more strongly bonded than wild males, increasing the level of interaction between them and with other pride members. The small territory size may also have caused the white lion males to spend more time with the pride, leading to more social interaction and cohesion than the wild males. The fact that one of the two adult lionesses was not genetically related to the other adult lioness, as would be typical within a wild pride, must also be considered. Further study is required to determine whether a white lioness would be established as a keystone female if integrated with wild tawny lionesses, and ultimately form a socially cohesive unit. Turner et al. (2015) [35] studied a pride of all white lions and observed that the pride formed a cohesive unit, hunting self-sufficiently and displaying similar behavioural interactions to wild lion prides, but unfortunately SNA was not used in that study. Historically, white lions have been observed as fully functional, socially connected members of wild tawny prides in the Timbavati Private Nature Reserve [28,30,34], and Kruger National Park [30,31,32], but this is the first time that SNA has been applied to better understand the social dynamics of white lions and to assess whether their social interactions are any different to those of wild tawny lions. Future studies should consider the limitations of this study by increasing the sample size, the territory size, and the number of male and female white lions in the pride to approximate that of wild lion prides. An increased sample size and territory size would mean a more natural ecosystem with more than one pride, so that the effects of territoriality and intraspectific competition on social behaviour can be more fully studied.

## 5. Conclusions

Social network analysis (SNA) showed that integrating white lion males of captive-origin with wild translocated tawny lionesses can form a socially cohesive and highly connected pride. Importantly, the white lion males displayed many of the expected social behaviours that have been observed in wild tawny lions. That a socially cohesive pride was observed is important, as this pride included individuals of differing backgrounds and was constructed by human intervention. This provides valuable knowledge on the social interactions of translocated lions, which may be relevant to the metapopulation management approach occurring in South Africa. Such information will become more pertinent if wild populations continue to decline and become more fragmented due to increasing anthropic pressure, resulting in the species being conserved within smaller fenced reserves.

## Figures and Tables

**Figure 1 animals-12-01985-f001:**
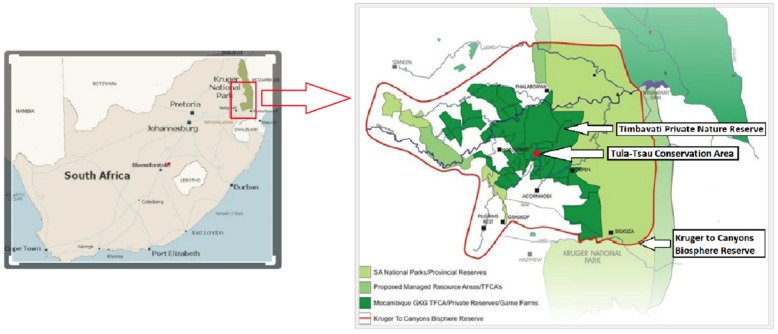
Location of the Tula-Tsau Conservation Area, Kruger to Canyons Biosphere Reserve, South Africa.

**Figure 2 animals-12-01985-f002:**
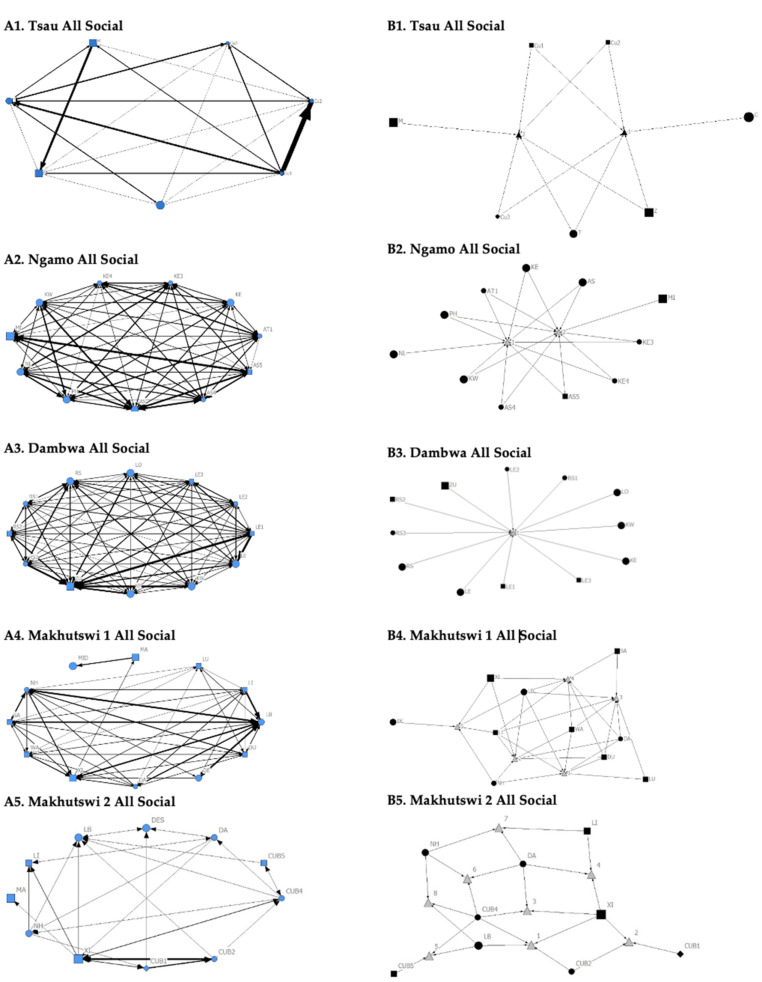
Sociogram (**A1**–**A5**) and clique (**B1**–**B5**) matrices of all social interactions, per pride. For both matrix types, squares (male), circles (female) and diamond (unknown sex) are nodes, representing individual lions, with node size indicating lion age (larger the symbol, the older the lion). For the sociograms, line thickness between dyads represents the strength of the association between lions. For the cliques, triangles signify a clique.

**Figure 3 animals-12-01985-f003:**
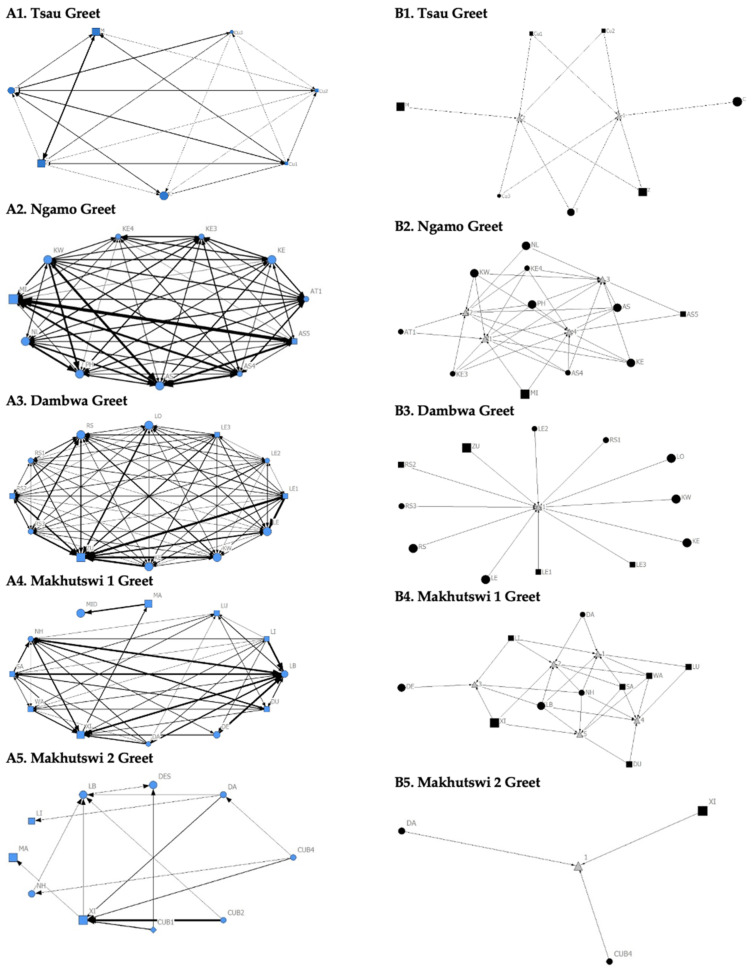
Sociogram (**A1**–**A5**) and clique (**B1**–**B5**) matrices of greet interactions, per pride. For both matrix types, squares (male), circles (female) and diamond (unknown sex) are nodes, representing individual lions, with node size indicating lion age (larger the symbol, the older the lion). For the sociograms, line thickness between dyads represents the strength of the association between lions. For the cliques, triangles signify a clique.

**Figure 4 animals-12-01985-f004:**
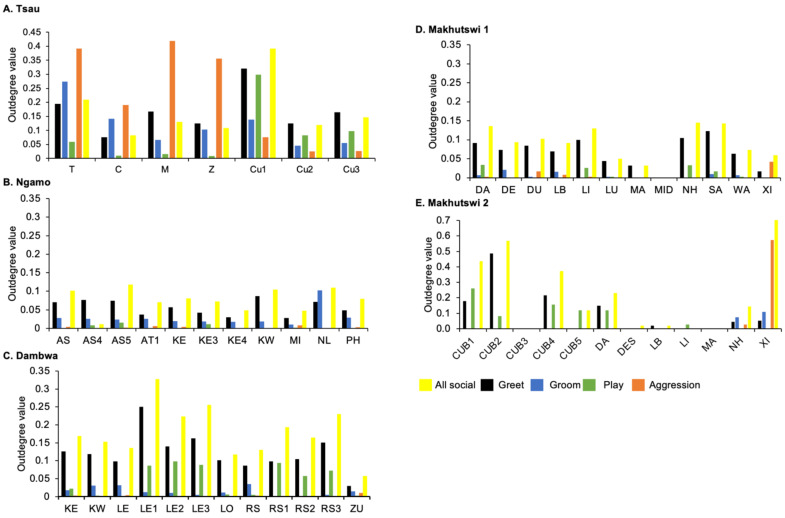
Normalised outdegree per social network (all social, greet, groom, play and aggression), per lion, per pride.

**Figure 5 animals-12-01985-f005:**
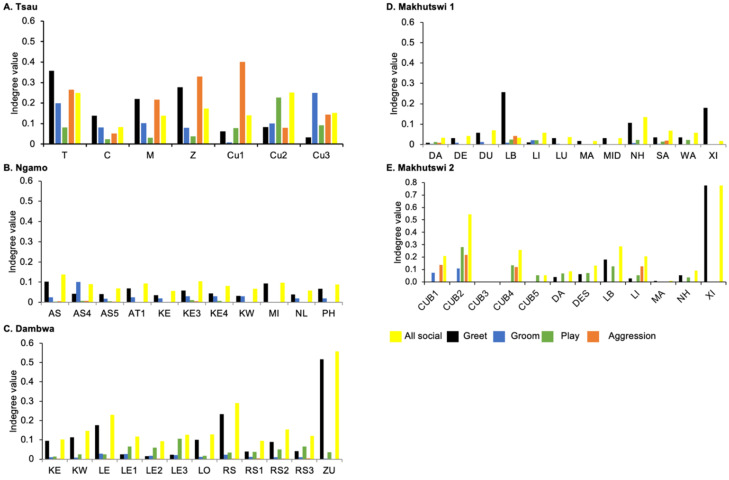
Normalised indegree per social network (all social, greet, groom, play and aggression), per lion, per pride.

**Figure 6 animals-12-01985-f006:**
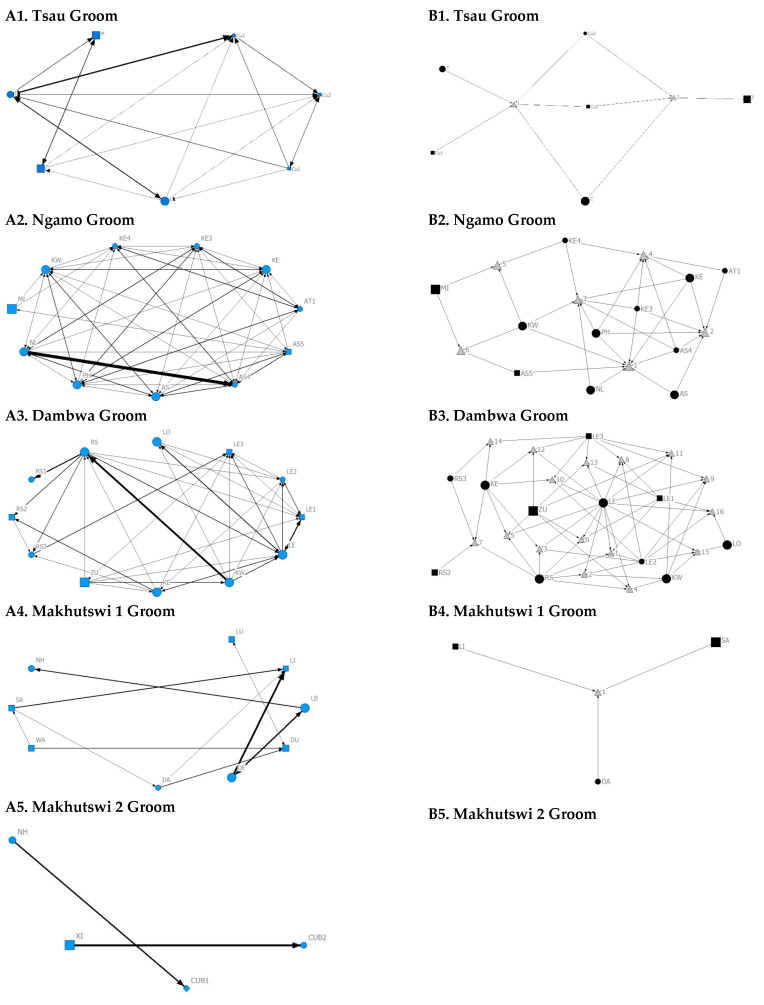
Sociogram (**A1**–**A5**) and clique (**B1**–**B5**) matrices of groom interactions, per pride. For both matrix types, squares (male), circles (female) and diamond (unknown sex) are nodes, representing individual lions, with node size indicating lion age (larger the symbol, the older the lion). For the sociograms, line thickness between dyads represents the strength of the association between lions. For the cliques, triangles signify a clique. Note: for B5. there was no clique due to insufficient interactions.

**Figure 7 animals-12-01985-f007:**
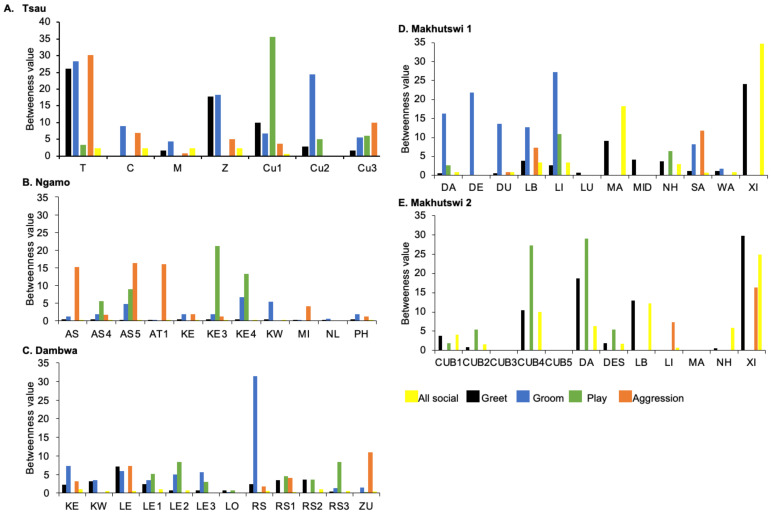
Normalised betweenness centrality per social network (all social, greet, groom, play and aggression), per lion, per pride.

**Figure 8 animals-12-01985-f008:**
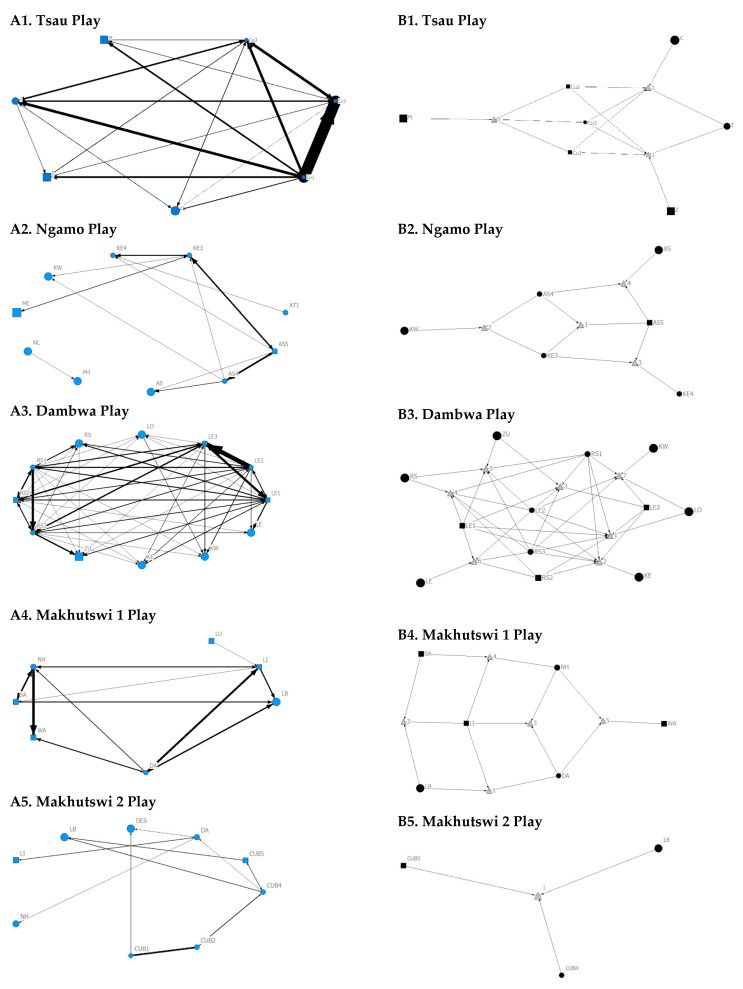
Sociogram (**A1**–**A5**) and clique (**B1**–**B5**) matrices of play interactions, per pride. For both matrix types, square (male), circles (female) and diamond (unknown sex) are nodes, representing individual lions, with node size indicating lion age (larger the symbol, the older the lion). For the sociograms, line thickness between dyads represents the strength of the association between lions. For the cliques, triangles signify a clique.

**Figure 9 animals-12-01985-f009:**
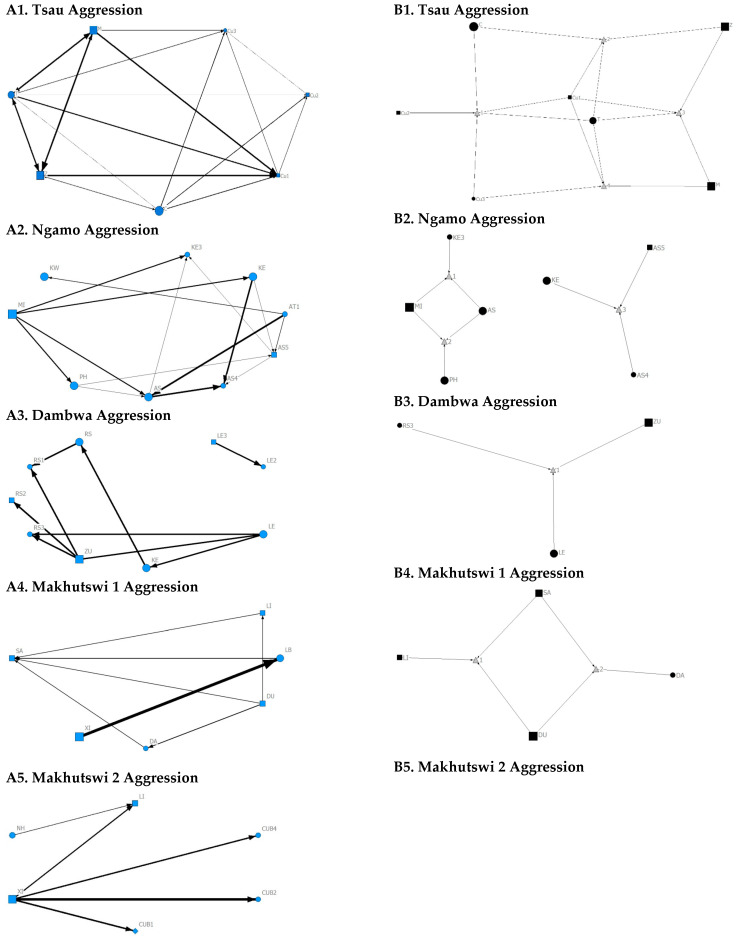
Sociogram (**A1**–**A5**) and clique (**B1**–**B5**) matrices of aggression interactions, per pride. For both matrix types, square (male), circles (female) and diamond (unknown sex) are nodes, representing individual lions, with node size indicating lion age (large the symbol, the older the lion). For the sociograms, line thickness between dyads represents the strength of the association between lions. For the cliques, triangles signify a clique. Note: for B5. there was no clique due to insufficient interactions.

**Table 1 animals-12-01985-t001:** Pride composition, GPS coordinates, reserve size and observational period of each pride.

	Adult (>4 Years)		Sub Adult (2–4 Years)		Cub (<2 Years)		Total	GPS Coordinates	Reserve Size	Observation Period
	Male	Female	Male	Female	Male	Female				
Tsau	2 *	2	0	0	2	1	7	24°23′ S	700 ha	7 April–10 May 2019
								31°08′ E		
Dambwa	1 *	6 *	0	0	3	3	13	17°50′ S	286 ha	21 September–6 December 2014
								25°45″ E		
Ngamo	1 *	5 *	1	4	0	0	11	19°30′ S	163 ha	14 May–21 June 2014
								29°44′ E		
Makhutswi 1	2	3	4	1	1	1	12	29°09′ S	25,000 ha	1 July–12 September 2014
								30°32′ E		
Makhutswi 2	2	2	0	1	2	3	10	29°09′ S	25,000 ha	17 August–20 October 2015
								30°32′ E		

* Indicates an individual who is of captive-bred origin.

**Table 2 animals-12-01985-t002:** Density values for each network, per pride, with a value of 0 indicating an unconnected pride and 1 indicating a highly connected pride.

Pride	Greet	Groom	Play	Aggression	All Social	Pride Composition
Tsau	0.643	0.476	0.643	0.595	0.929	1
Ngamo	0.891	0.664	0.145	0.136	0.964	1
Dambwa	0.78	0.303	0.515	0.515	0.939	1
Makhutswi 1	0.394	0.091	0.114	0.114	0.492	1
Makhutswi 2	0.106	0.015	0.091	0.038	0.22	1

## Data Availability

This research and its data forms part of a PhD thesis registered with Leiden University, Leiden, Netherlands.

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
