# Peer review of "Assessing the Social Cohesion of a Translocated Pride of White Lions Integrated with Wild Tawny Lions in South Africa, Using Social Network Analysis"

_animals, 2022, doi:10.3390/ani12151985_

Round 1
Reviewer 1 Report
This seems to be a sound paper, with some very interesting content, my main contents are with regards to the statistical elements of the paper.
First of all, the p-values are in general quite high in many of the cases. A publication by the American Statistical Association in 2019 insisted scientists stop using p < 0.05 as an indicator of "statistical significance", considering this p-value to have an associated probability of being a Type I statistical error of 28.9%. Following this, some authors have proposed p < 0.005 or p < 0.003 to be more robust. From this perspective, phrases such as "A significant positive association was found for the Tsau pride between the groom..." on lines 367-368 should be changed to "A positive association", so as to not make too strong a conclusion from a very high p-value (0.0384 or 0.0393). I suggest revising in general the text to check for possible points that could be considered conflictive.
Consider:
Benjamin, D. J., Berger, J. O., 2019. Three recommendations for improving the use of p-values. Am. stat. 73, 186–191.
Colquhoun, D., 2019. The False Positive Risk: A proposal concerning what to do about p-values. The American Statictician 73, 192–201.
Wasserstein, R. L., Schirm, A. L., Lazar, N. A., 2019. Moving to a world beyond “p < 0.05”. Am. Stat. 73, 1–19. https://doi.org/10.1080/00031305.2019.1583913
Looking at the results in more detail, another point I notice is the mention on a number of occasions on the "lack" or "presence" of differences/correlations , yet no statistical values are reported to support this observation. Example: "There was no significant association between any of the interaction networks...", "a correlation was found between the all social and greet matrices for all prides...", "Spearman's correlation found no association between greet indegree and greet outdegree...", "Kendall's tau analysis found no association between greet centrality and any of the...". Please complete these phrases with the relevant information so as to assess these "lack of differences" or "correlations" etc.
With regards to Table 1 - I am not sure I understand the meaning/purpose/importance of the "Pride Composition" column.
Finally, out of interest, why did the authors consider a social interaction to have ended after 1 minute? What are the criteria to select this interval?
Reviewer 2 Report
Social cohesion is an important factor determining the success of lion translocation/reintroduction; SNA has proved a useful method to assess the structure of animal societies. Applying the SNA method, this manuscript demonstrated that, for the first time, reintroduced captive-origin white lions integrated into a pride can form a socially cohesive pride. The results are very pertinent to the increase of lion survivorship with mitigated anthropogenic impact in South Africa. Considering the novelty and significancy of the manuscript, the reviewer recommends publication of the manuscript after some revisions regarding the following aspects:
1. Section Materials and Methods: (1) proving a zoomed-in map of the study site will make it easier to understand to location; (2) adding a diagramme which can show the overall design of the study will help the audience understand better the underlying research ideas.
2. Section Results: in Figures 1, 2, 5, 7, and 8, would it be possible to increase the font size of the code of each node; the current codes are a bit blurry.
3. Section Discussion: please elaborate a bit on the effect of sample size and territory size (cf. the ending lines of this section); for instance, how the results would be given increased/decreased population density.
